# Structural Property, Immunoreactivity and Gastric Digestion Characteristics of Glycated Parvalbumin from Mandarin Fish (*Siniperca chuaisi*) during Microwave-Assisted Maillard Reaction

**DOI:** 10.3390/foods12010052

**Published:** 2022-12-22

**Authors:** Jingjing Tai, Dan Qiao, Xue Huang, Huang Hu, Wanzheng Li, Xinle Liang, Fuming Zhang, Yanbin Lu, Hong Zhang

**Affiliations:** 1School of Food Science and Biotechnology, Zhejiang Gongshang University, Hangzhou 310018, China; 2Department of Chemical and Biological Engineering, Center for Biotechnology and Interdisciplinary Studies, Rensselaer Polytechnic Institute, Troy, NY 12180, USA; 3Collaborative Innovation Center of Seafood Deep Processing, Key Laboratory of Aquatic Products Processing of Zhejiang Province, Institute of Seafood, Zhejiang Gongshang University, Hangzhou 310012, China

**Keywords:** parvalbumin, Maillard reaction, glycation site, structural and immunological properties, in vitro

## Abstract

This study was aimed to investigate the structural and immunological properties of parvalbumin from mandarin fish during the Maillard reaction. The microwave-assisted the Maillard reaction was optimized by orthogonal designed experiments. The results showed that the type of sugar and heating time had a significant effect on the Maillard reaction (*p* < 0.05). The SDS-PAGE analysis displayed that the molecular weight of parvalbumin in mandarin fish changed after being glycated with the Maillard reaction. The glycated parvalbumin was analyzed by Nano-LC-MS/MS and eleven glycation sites as well as five glycation groups were identified. By using the indirect competitive ELISA method, it was found that microwave heating gave a higher desensitization ability of mandarin fish parvalbumin than induction cooker did. In vitro gastric digestion experiments showed that microwave-heated parvalbumin was proved to be digested more easily than that cooked by induction cookers. The microwave-assisted Maillard reaction modified the structure of parvalbumin and reduced the immunoreactivity of parvalbumin of mandarin fish.

## 1. Introduction

Mandarin fish (*Siniperca chuaisi*), also known as Osmanthus fish, is a popular fishery product with characteristics of delicious taste, fewer bones and of rich nutrition [1]. However, fish proteins, one of the eight allergenic foods, can cause IgE-mediated allergic symptoms, such as pruritus and red rashes [2], which hinder the intake of quality proteins of mandarin fish and other fishes. The major fish allergen has been identified as parvalbumin, with 92.5% of fish allergy individuals having specific IgE reaction with parvalbumin [3]. Parvalbumin, a calcium-binding albumin protein, has a molecular weight around 12 kDa and isoelectric point (pI) of 4.0–5.5 [4]. The reports show that the parvalbumin mainly exists in the fast skeletal muscle of white meat [5], each fish (codfish, chimaeras and stingray) containing two to three subtypes at the same time, while some teleost fishes (gadus morhua, salmon salar and anguilla japonica) can produce 3–5 subtypes during their entire growth and development [6]. The conserved sequences of allergenicity parvalbumin form different fish species have been reported [3]. Currently, more and more studies have focused on the allergen of mandarin fish, which is an important economic species of freshwater fish, widely distributed in China [7]. Yang et al. (2012) proved that parvalbumin is one of the allergens of mandarin fish, and completed the gene cloning and sequence analysis of it [8]. Kuehn et al. (2017) investigated the existence and content of parvalbumin in mandarin fish, and established a qualitative and quantitative detection method for parvalbumin of mandarin fish [9]. In recent years, there are many reports on reducing food allergy using biological and physical methods. Li et al. (2016) used the hydrolysis method to eliminate the allergenicity of wheat [10]. Some researchers also used high-intensity ultrasound to significantly reduce the allergenicity of shrimp tropomyosin [11]. However, the resistance of parvalbumin to high temperature and enzyme results in little effect of these methods on its desensitization [12]. Considering the materials and reaction conditions, these reported methods also have many limitations for the application in daily cooking.

The Maillard reaction, also known as glycation, is a non-enzymatic browning reaction between proteins and reducing sugars [13]. It plays an important role in food processing owing to its incredible advantages in changing food qualities such as color, flavor and nutritional value [14]. Recently, an obvious trend arose to use the Maillard reaction to reduce protein-induced food allergy. It was reported that soybean proteins were glycated by fructose with the Maillard reaction to decrease their allergenicity [15]. Fu et al. (2019) confirmed that the allergenicity of shrimp tropomyosin could be significantly reduced by the Maillard reaction with different reducing sugars [16]. Han et al. (2018) demonstrated that the Maillard reaction could reduce the sensitization of tropomyosin (TM) and arginine kinase from *Scylla paramamosain* [17]. Therefore, the Maillard reaction has proved to be a promising method to reduce the immunoreactivity [18].

Most of the studied materials of the Maillard reaction in these reports were purified proteins, protein hydrolysates or recombinant proteins, and some of the reactions lasted for several days [19,20]. All these conditions are obviously not suitable for daily cooking. Microwave cooking, known for its fast heating and convenient operation, has becoming a common choice of daily cooking. In the current study, the mandarin fish was used in the Maillard reaction to explore the desensitization of its parvalbumin using the microwave heating method. The main goal of this study was investigation how the Maillard reaction affects the structural and immunological properties of parvalbumin and its gastric digestibility. The optimum conditions of the microwave-assisted Maillard reaction were determined by orthogonal designed experiments. The glycated sites and derived groups of parvalbumin were further analyzed by nano-liquid chromatography-tandem mass spectrometry (Nano-LC-MS/MS). The relationship between structural modification of parvalbumin by the Maillard reaction and its immunoreactivity was also investigated using an indirect competitive ELISA assay.

## 2. Material and Methods

### 2.1. Materials

Mandarin fish *(Siniperca chuaisi)* were bought in a Yonghui supermarket (Hangzhou, China). Pepsin and bicinchoninic acid (BCA) protein assay kits were purchased from Sigma (St. Louis, MO, USA). Anti-rabbit IgG peroxidase-conjugated antibody produced in goat was purchased from Polila (Beijing, China). 3,3′,5,5′-Tetramethylbenzidine (TMB) was purchased from Aladdin (Shanghai, China). Formic acid (FA) and acetonitrile (ACN) (MS grade) were bought from Thermo Fisher Scientific (Waltham, MA, USA). D-glucose, maltose, sucrose and other chemicals were purchased from Bio-Rad in the best available purity grade.

### 2.2. Orthogonal Designed Experiments of Microwave-Assisted Maillard Reaction

#### 2.2.1. Sample Preparation

The mandarin fish samples were transported back to the laboratory at 4 ± 1 °C. The fish flesh was separated from the fat layer and internal organs. Then, it was washed three times with distilled water and smashed using an ultramicro pulverizer. Finally, 20 g of fish flesh was accurately weighed, packed and stored at −80 °C until further analysis.

#### 2.2.2. Orthogonal Designed Experiment

To reduce the number of experiments, the orthogonal designed experiment approach was applied. In accordance with the sugar commonly used in household and food industry production, glucose, maltose and sucrose were selected as a first factor (A). Beside the type of sugar, according to the results of the pre-experiment, sugar concentration (0.5, 1, 1.5 mol/L) (B) and microwave heating time (5, 10, 15 min) (C) were selected as additional two factors in the orthogonal designed experiments. Nine experimental conditions were designed: G1 (glucose, 0.5 mol/L, 5 min), G2 (glucose, 1 mol/L, 10 min), G3 (glucose, 1.5 mol/L, 15 min), M1 (maltose, 0.5 mol/L, 10 min), M2 (maltose, 1 mol/L, 15 min), M3 (maltose, 1.5 mol/L, 5 min), S1 (sucrose, 0.5 mol/L, 15 min), S2 (sucrose, 1 mol/L, 5 min) and S3 (sucrose, 1.5 mol/L, 10 min). Another three experimental conditions were selected as a control: G-Heating (glucose, 1.5 mol/L, 15 min), M-Heating (maltose, 1.5 mol/L, 15 min), S-Heating (sucrose, 1.5 mol/L, 15 min). The high flame heating (2450 MHz) was selected as a condition during the microwave-assisted Maillard reaction, while 800 W was selected during the induction cooker-assisted Maillard reaction (control). In all experiments, the material-liquid ratio (*w*/*v*) was 1:4.

#### 2.2.3. Preparation of Crude Extract

After heating experiments, 20 g reaction mixture was rigorously homogenized with 160 mL buffer solution (50 mM K_2_HPO_4_, 50 mM KH_2_PO_4_, pH 7.5) and rotated at 4 °C overnight. Then, the homogenate was centrifuged at 10,000 g for 10 min at 4 °C, and the supernatant was collected as crude protein. The protein concentration in crude extract was determined by the Bradford method using bovine serum albumin as a standard. The final concentration of protein was adjusted to 2 mg/mL by buffer for further analysis.

#### 2.2.4. Determination of Browning Index of Maillard Reaction

The browning index as an indicator of the degree of the Maillard reaction can be estimated by the measurement of absorbance at 420 nm. Absorbance at 420 nm in the diluted samples of crude extracts (200 µL) was measured using a microplate plate reader (SuPerMax, Shanghai, China) [21].

### 2.3. Purification of Parvalbumin

Each diluted crude extract sample was mixed with 60% volumes of saturated ammonium sulfate solution. After standing for 2 h, the mix solution was centrifuged at 10,000 rpm/min for 20 min. The supernatant, a partially purified paralbumin fraction, was collected and stored at −20 °C until analysis.

### 2.4. SDS-PAGE (Sodium Dodecyl Sulfate–Polyacrylamide Gel Electrophoresis) Analysis

Each purified parvalbumin sample was boiled in 4 × protein SDS-PAGE loading buffer with a final concentration of 1 mg/mL for 5 min 95 °C. Gels were 15% acrylamide gel and electrophoresis was conducted at a constant voltage of 100–140 V for 2–3 h and stained with Coomassie Brilliant Blue [22]. The molecular weight of parvalbumin in the sample was estimated by using prestained protein standards (6.5 to 200 kDa, Taraka, Beijing, China).

### 2.5. Nano-LC-MS/MS Analysis of Glycated Parvalbumin Peptides

After in-gel tryptic digestion [23], the glycated parvalbumin peptide samples were analyzed and identified by a Nano-LC-MS/MS protocol. An Eksigent ChromXP trap column (100 μm × 3 cm, 3.0 μm 150 Å, C18) has been used to detect the peptides obtained by enzymatic hydrolysis, which were then separated by a Eksigent ChromXP analytical column (75 μm × 15 cm, 3.0 μm, 120 Å, C18) and then with Triple TOF 5600 mass spectrometer from AB SCIEX (Applied Biosystems, Foster City, CA). The 5% ACN/0.1% FA in water and 5% water/0.1% FA in ACN were used as mobile phase A and B, respectively. Gradient elution conditions are as follows: 0–25 min, 5–40% B. The injection volume and elution flow rate were 3 μL and 300 nL/min, respectively. A Nano-TurboIon Spray™ interface and electron spray ionization (ESI) operating in the positive mode were used in MS/MS detection. The ESI factors were as follows: Ion spray voltage (IS), 2.3 kV; Ion source gas1 (GS1), 30 psi; Ion source gas2 (GS2), 5 psi; Curtain gas (CUR), 25 psi; Ion source temperature, 150 °C. The mass data were scanned in IDA mode (Information Dependent Acquisition). All instrument control, data acquisition and the processing were performed using the associate Analyst 1.7.1 software (SCIEX, Framingham, MA, USA). The amino acid sequence of eluted peptides was further processed and analyzed manually using PEAKS Studio 8.5 software (Bioinformatics Solutions Inc., Waterloo, ON, Canada) [24], protein database in fasta format was download from the UniProt KB database (https://sparql.uniprot.org) accessed on 1 October 2021.

### 2.6. Immunization of Rabbit by Mandarin Fish Parvalbumin and Preparation of Antiserum

Polyclonal antibodies were obtained in New Zealand white rabbits (2.5 kg, female, provided by the Animal Experimental Center of Hangzhou Normal University, Hangzhou) by injection of purified parvalbumin, complied with Shibahara’s research [25]. Specific conditions are as follows: 2 mg/mL purified PV (purity higher than 85%) was mixed with Freund’s incomplete adjuvant 1:1, and five New Zealand white rabbits were intramuscularly injected with 0.5 mg/kg body weight. Each rabbit was injected once every two weeks, a total of four times.

### 2.7. Indirect Competitive Enzyme Linked Immunosorbent Assay (ic-ELISA)

ELISA was performed in 96-well microplate as Tukiran described [26]. The plate was coated with 100 μL/well parvalbumin (2 μg/mL) at 4 °C overnight. After washing three times with washing buffer (0.05% Tween 20 in PBS), the plate was blocked with 200 μL/well of blocking buffer (5% Non-fat milk powder in PBS) at 37 °C for 1.5 h. Each sample of parvalbumin in G3, M2, S1, G-Heating, M-Heating, S-Heating, Native was 1:1 mixed with antiserum (1: 4000) as premix PV-antibodies, respectively. The different premix PV-antibodies were added into plate and incubated at 37 °C for 1 h. After washing three times with washing buffer, the diluted goat-antirabbit immunoglobulin conjugated to horseradish peroxidase (HPR) (dilution 1:5000) was added 100 μL into each well and incubated at 37 °C for 1.5 h. After a final washing step, 100 μL/well of substrate solution of TMB was added. The color reaction was stopped after 30 min by adding 100 μL/well of 0.05 M H_2_SO_4_. The absorbance was measured at 450 nm. The binding capacity between sample and IgG was calculated as follow formula:
$$\mathrm{Inhibition}\;\left(\%\right)\;=\;1\;-\;\frac{\left(A_{s}-A_{\min}\right)}{\left(A_{\max}-A_{\min}\right)}\times 100\%$$
where *A**_max_* is the absorbance in non-competition system by containing polyclonal antibodies and *A**_min_* is the absorbance of using PBS, which is also the absorbance in the competitive system [27].

### 2.8. In Vitro Gastric Digestion

Simulated gastric digestion of purified parvalbumin samples after the heating experiment was performed according to a modified method by Thomas [28]. A single tube containing 1.52 mL of simulated gastric fluid (SGF; 0.084 N HCl, 35 mM NaCl, pH 1.2 or 2.0, and 4000 U of pepsin) was preheated to approximately 37 °C prior to the addition of 0.08 mL of 2 mg/mL purified parvalbumin samples. The tube contents were mixed by mild vortexing and the tube was immediately placed in water bath to incubate at 37 °C for 60 min. At time point 0.5, 1, 2, 5, 10, 30 and 60 min of initiation of incubation, 200 µL of digestion mixture was removed and mixing with 70 μL of 500 mM Na_2_CO_3_ (pH 11) and 4 × protein SDS-PAGE loading buffer to stop enzyme reaction. After that, samples were heated at 95 °C for 10 min and analyzed by SDS-PAGE.

### 2.9. Statistical Analysis

Data were expressed as mean ± standard deviation in triplicate. Statistical analysis was carried out by GraphPad Prism software 4.0. (GraphPad, San Diego, CA, USA). Statistical differences under different levels as ** (*p* < 0.01) or * (*p* < 0.05) were considered statistically significant. Protein alignment and sequence analyses were performed by using PEAKS Studio 10.6 (Bioinformatics Solutions Inc., Waterloo, ON, Canada). Peakview 1.2 (SCIEX, Framingham, MA, USA) was used for output of mass spectrum data.

## 3. Results and Discussion

### 3.1. Optimization of the Maillard Reaction

The browning index was used to indicate the degree of the Maillard reaction, which was evaluated as the absorbance value at 420 nm. The higher absorbance value means a higher degree of the Maillard reaction. As shown in Table 1, according to the type of sugar tested in the orthogonal experiment, the highest absorbance value was G3 (glucose, 1.5 mol/L, 15 min), followed by M2 (maltose, 1 mol/L, 15 min) and S1 (sucrose, 0.5 mol/L, 15 min). The type of sugar and heating time by microwave heating had significant influences (*p* < 0.05) on the Maillard reaction, while the substrate concentration had no significant influence (*p* > 0.05). The order of three influencing factors was: type of sugar > heating time > substrate concentration. It is noteworthy that glucose brought maximum degree of browning, followed by maltose and sucrose. The reason was that glucose and maltose are free carbonyl groups, which can react with protein immediately. Sucrose is not a reducing sugar with a free carbonyl group and the activation of this group can be achieved only through degradation steps of sugar [29]. Therefore, the browning index with sucrose was lower than the other two sugars.

From Table 1, the values of the browning index obtained during microwave heating was significantly higher compared to the values obtained using an induction cooker, when compared to G3/G-Heating, M2/M-Heating and S1/S-Heating. Under the condition of the same type of sugar, concentration and heating time, microwave heating gave a higher yield of the Maillard reaction rate and browning degree than induction heating. Using color as a criterion for evaluating the Maillard reaction, it was found that the products from G3 and M2 samples seemed obvious yellow or brown, while the products from G2 and S1 samples looked light yellow. The change of color in other samples could not be observed by the naked eye, especially in the control group heated by an induction cooker.

Interestingly, our results indicating that sugar type and heating time have a statistically significant effect on the Maillard reaction (*p* < 0.05), and the sugar concentration has no significant effect (*p* > 0.05), which was slightly different from the results of previous reports [20]. The main reason is the different types of sugar and raw material states used in the Maillard reaction as well as conditions of heating. In our study, microwave heating was applied, which is obviously different from traditional heating methods. Microwave heating can not only speed up chemical synthesis reaction, but also shorten reaction time from several days to several minutes [30]. As for substrate concentration, the levels in our experiment were greater than that previous [20]. In general, the results showed that the type of sugar and heating time had a significant effect on the Maillard reaction.

### 3.2. SDS-PAGE Analysis of Parvalbumin

The glycated parvalbumin from the Maillard reaction were further analyzed by SDS-PAGE. By analyzing the change of parvalbumin molecular weight, it can reflect the degree of the Maillard reaction. As shown in Figure 1b, compared with molecular weight of native mandarin fish parvalbumin, the glycated parvalbumin from M2 (maltose, 1 mol/L, 15 min) gave the shortest molecular weight migration, followed by G3 (glucose, 1.5 mol/L, 15 min). The different migration bands were confirmed by the Quantity One software (Bio-Rad, Hercules, CA, USA). SDS-PAGE also indicated some samples were degraded into low molecular weight fragments, such as G1 (glucose, 0.5 mol/L, 5 min), G2 (glucose, 1.0 mol/L, 10 min) and M1 (maltose, 0.5 mol/L, 10 min) (Figure 1a), which is similar to the SDS-PAGE patterns of Teodorowicz et al. (2013) [22]. The Maillard reaction led to protein degradation, while glycation increased the parvalbumin molecular weight. These two states occur simultaneously and affect the molecule weight of glycated parvalbumin final products [31]. In the early stage of the Maillard reaction, protein degradation was the dominant reaction. The SDS-PAGE showed the reduced molecular weight of parvalbumin under the reaction condition of G1, G2 and M1. With the progression of the Maillard reaction, the glycation played a dominant role and gave an increasing on protein molecular weight. Therefore, the reduced molecular weight from protein degradation could be compensated by glycation from the Maillard reaction. The sample G3 showed the shortest migration (highest MW) compared to G1 and G2 on SDS-PAGE, indicating the most completed Maillard reaction occurred under G3 conditions, which is consistent with the results of orthogonal experiments.

The effects of different heating methods on the Maillard reaction showed the migration distance of the G3 product is shorter than the samples from G-Heating, in comparison with band of native parvalbumin (Figure 1b). The similar pattern was obtained for samples M2 and M-Heating, indicating that microwave heating produced a higher reaction yield than an induction cooker in the Maillard reaction with glucose and maltose. However, compared with native protein, the products from S1 and S-Heating showed no obvious difference in migration on SDS-PAGE. When sucrose was used, the different heating methods had no significant effect on the degree of the Maillard reaction.

### 3.3. Identification of Glycation Sites in Parvalbumin

To investigate the glycation of parvalbumin, samples G3 and M2 with the best yield of the Maillard reaction were chosen to be analyzed by Nano-LC-MS/MS. Native mandarin fish parvalbumin was also analyzed as control. Amino acid analysis of tryptic peptides, obtained from native parvalbumin and Uniprot database matching, showed two isoforms of parvalbumin, named PV-I (99% coverage) and PV-II (75% coverage), respectively (data shown as Figure 2).

PV-I and PV-II are the most common parvalbumin isoforms, and their protein structure are similar; both of them contain 108~109 amino acids, but the amino acid sequence is different, and the homology of sequence is only 50 to 80%. Both isoforms can cause allergic reactions, but PV-I has stronger binding ability with IgG and IgE than PV-II [32]. The mass spectrums of samples from G3 and M2 were manually analyzed and compared with native parvalbumin by PEAKS Studio (BSI, Waterloo, ON, Canada). The possible modified groups, including formyl [33], carboxymethyl, carboxyethyl [34], pyruvaldehyde, pyrraline [35], glucose, maltose [36], arginine and lysine [37], were analyzed. The modified groups, including formyl (Frm), carboxymethyl (Cmc), glucose (G), arginine [18] and the sulfate group * (M = 97.03), were found in the real samples. In this study, five potentially glycated peptides and sites (PV-I peptides 34–45, 46–55, 77–88 and 89–108 and PV-II peptide 77–88) were identified through molecular mass comparison with the theoretical tryptic peptide masses in Table 2 (the corresponding mass spectrum were shown in Appendix A). A total of six peptides from G3 were glycated, and three peptides were modified in M2, which indicated that glucose was more susceptible to the Maillard reaction than maltose. The modified groups caused by the Maillard reaction were mainly formylation and carboxymethylation, which was consistent with Laoque’s report [38].

The modified groups and sites may change the linear and conformational epitope of the parvalbumin, which could affect its sensitization ability. Glycated K45 and K55 residues located with the three tetrapeptides (DEIK, DQDK and DELK, red, see Figure 3) corresponded to the reported epitopes of allergen M Gad c1 [39]. Thus, the glycated sites K45 and K55 could hinder the binding of parvalbumin epitope to IgE. The glycated sites T_79_ and D_80_ residues in PV-II of M2 were consistent with the stimulated epitope of cod allergy peptide (blue) matched by Untersmayr et al. (2006) [40]. Moreover, the glycated K_39_ residue in PV-I of G3 and M2 was adjacent to the blue region, which may also affect the binding of linear epitope of parvalbumin to IgE. The gray marker was the calcium binding site of two subtypes parvalbumin found by Marchler-Baure et al. (2015) [41], which is highly conserved and maintains the stereo domains of epitope of parvalbumin in various fishes [42]. K_97_ residue in peptide 89–108 of PV-I overlapped with the highly conserved Ca^2+^ binding regions 52–63, 91–102, suggesting a direct blocking of IgE-binding epitopes by the glycation. Overall, eleven glycation sites and five glycation groups of parvalbumin through the Maillard reaction were identified by Nano-LC-MS/MS. Glycation resulted from the Maillard reaction may destroy the conserved structure and the predicated epitopes in parvalbumin, which can further affect the immunogenicity and allergenicity of the parvalbumin [43].

### 3.4. The Influence of Glycation on Parvalbumin Immunological Properties

The influence of parvalbumin immunological properties from the Maillard reaction was further assessed using ic-ELISA. The lower immunoreactivity was indicated by the lower inhibition rate in the ic-ELISA tests. Compared with the native parvalbumin (Figure 4), the inhibition rates are all samples obtained when both types of heating (microwave or induction cooker) were lower, indicating that the Maillard reaction offered some effect on reducing the antigenicity of the parvalbumin. When compare the inhibition rates of different sugars (samples G3, M2 and S1) to the inhibition rate of native parvalbumin, it was found that glucose had the highest inhibition effect, up to 66.03%, followed by maltose and sucrose, suggesting glucose was the best sugar substrate for the Maillard reaction. These findings were inconsistent with the results of orthogonal experiments. It was found that the inhibition rates of samples obtained during microwave heating were significantly lower than those obtained during induction cooker heating. These results suggest that the microwave cooking-assisted Maillard reaction could be a better way to reduce the immunogenicity and immunoreactivity of the parvalbumin than that by induction cooker heating. The mechanism of influence of glycation on the parvalbumin immunological properties could be the glycation from the Maillard reaction blocking the IgG/IgE-binding to the linear and conformational epitopes of parvalbumin [44].

### 3.5. In Vitro Gastric Digestion

Resistance of protein to pepsin digestion can be used as a marker of potential allergenicity [45]. The digestibility of glycated parvalbumin samples from different heating methods was verified by SDS-PAGE. As shown in Figure 5, the band of G3 parvalbumin completely disappeared within 2 min, while the product from G-Heating was completely hydrolyzed at 15 min, indicating that the digestibility rate of product from G-heating was lower than that from G3 heated by a microwave. Comparing the digestibility between products from M2 and M-heating, it showed that the protein bands of M2 were smeared after 2 min and completely disappeared at 60 min, while the band from the M-Heating product was still visible, indicating that the product from M2 was easier to be digested than that from M-Heating. Similar results were obtained in S1/S-heating. Thus, the glycated parvalbumin from the microwave-assisted Maillard reaction seems easier to be digested than the product from induction heating. On the other hand, comparing the glycated samples from G3, M2 and S1, the sample from G3 was the easiest one to be digested, suggesting that glucose was the best choice for the Maillard reaction. It is worth noting that the result of the rapid degradation of glycated parvalbumin by pepsin in our study was different from Zhao [46], who reported that glycation with the Maillard reaction could protect recombinant silver carp parvalbumin from degradation by pepsin. The main reason was that microwave heating can accelerate the reaction rate, thus reducing the exposure of hydrophobic groups of parvalbumin and making it easier to be digested [47].

Overall, the digestibility of the samples from the seven reaction conditions was in the following order (from easy to difficult): G3 > G-Heating > S1 > M2 > Native > M-Heating ≈ S-Heating. G3 was the optimum condition to produce glycated parvalbumin with high digestibility.

## 4. Conclusions

The structural and immunological properties of glycated mandarin fish parvalbumin from the microwave-assisted Maillard reaction were investigated in this work. Results from orthogonal designed experiments showed that the type of sugar and heating time had significant effect, while sugar concentration had no significant effect on the Maillard reaction. SDS-PAGE analysis proved that the molecular weight of parvalbumin changed after being glycated with the Maillard reaction. Eleven glycation sites and five glycation groups were identified from the glycated parvalbumin by Nano-LC-MS/MS. This suggests that as the Maillard reaction took place and the conserved structure and the predicated epitopes in parvalbumin may be destroyed and, thus, may affect the parvalbumin allergenicity. The ic-ELISA results showed the allergenicity of parvalbumin after the Maillard reaction was significantly reduced and gastric digestion results showed G3 (Glucose, 1.5 mol/L, 15 min) was the easiest one to be digested. Compared with different heating methods, microwave heating seems a better cooking method, as indicated by its product having reduced immunoreactivity and easy gastric digestion.

In conclusion, the microwave-assisted Maillard reaction proved to change the structure of parvalbumin and then reduce the immunoreactivity of parvalbumin of mandarin fish. We believe our current study provides some fundamental information for the food-processing industry or daily cooking to using microwave heating as a method to reduce the sensitized downside of certain food products.

## Figures and Tables

**Figure 1 foods-12-00052-f001:**
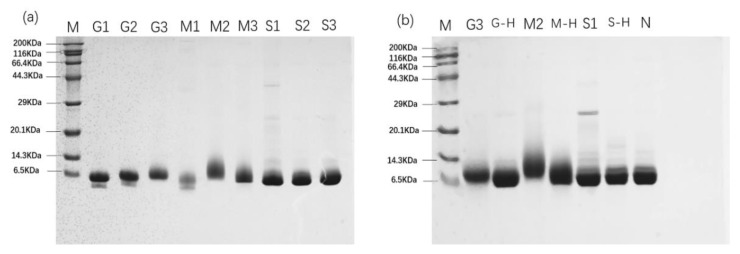
SDS-PAGE analysis of parvalbumin samples obtained during the Maillard reactions. (**a**) SDS-PAGE of parvalbumin from orthogonal experiments; M: MW Markers; G1 (Glucose, 0.5 mol/L, 5 min), G2 (Glucose, 1.0 mol/L, 10 min), G3 (Glucose, 1.5 mol/L, 15 min), M1 (Maltose, 0.5 mol/L, 10 min), M2 (Maltose, 1 mol/L, 15 min), M3 (Maltose, 1.5 mol/L, 5 min), S1 (Sucrose, 0.5 mol/L, 15 min), S2 (Sucrose, 1.0 mol/L, 5 min) and S3 (Sucrose, 1.5 mol/L, 10 min) were heated by a microwave; (**b**) SDS-PAGE analysis on native parvalbumin and glycated parvalbumin heated by a microwave or an induction cooker. N: native (unmodified and unheated) parvalbumin; G-Heating (Glucose, 1.5 mol/L, 15 min), M-Heating (Maltose, 1.5 mol/L, 15 min) and S-Heating (Sucrose, 1.5 mol/L, 15 min) were heated by an induction cooker.

**Figure 2 foods-12-00052-f002:**
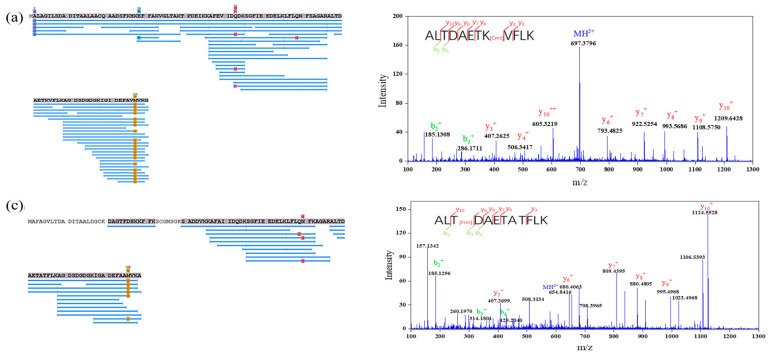
Peptide matching diagram of parvalbumin isoforms PV-I (**a**) and PV-II (**c**) in native sample and their mass spectrograms, (**b**) and (**d**), respectively.

**Figure 3 foods-12-00052-f003:**
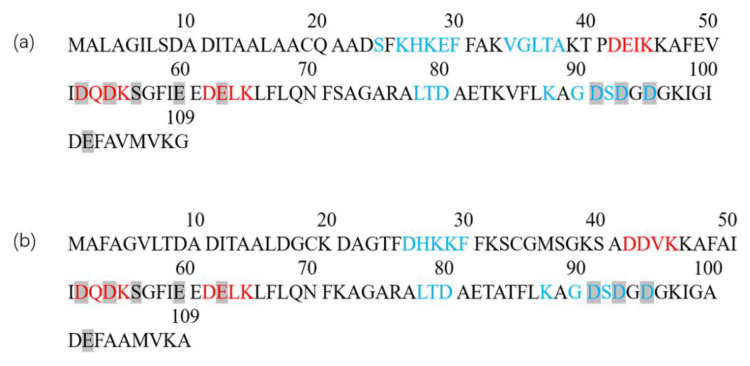
Amino acid sequence of parvalbumin isoforms PV-I (**a**) and PV-II (**b**) in mandarin fish. The amino acid coded in red was corresponded to the epitope of Baltic cod allergen M Gad c1. The amino acid coded in blue matched to the mimotope by phage display technique. The gray shaded section was a calcium-binding site found through the UniProt website.

**Figure 4 foods-12-00052-f004:**
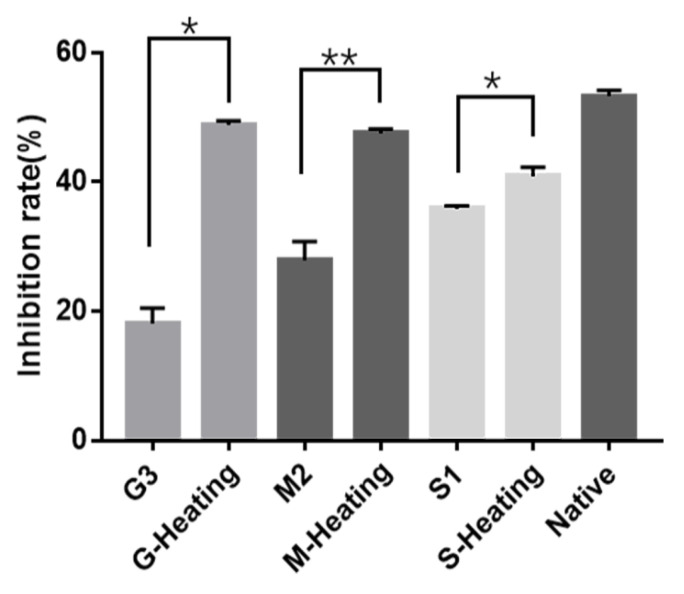
The inhibition of native parvalbumin and glycated parvalbumin heated by a microwave or an induction cooker. G3 (Glucose,1.5 mol/L,15 min), M2 (Maltose,1 mol/L,15 min) and S1 (Sucrose, 0.5 mol/L, 15 min) were heated by a microwave; G-Heating (Glucose, 1.5 mol/L, 15 min), M-Heating (Maltose, 1.5 mol/L, 15 min) and S-Heating (Sucrose, 1.5 mol/L, 15 min) were heated by an induction cooker. Native was unheated parvalbumin. * means (*p* < 0.05) or ** means (*p* < 0.01).

**Figure 5 foods-12-00052-f005:**
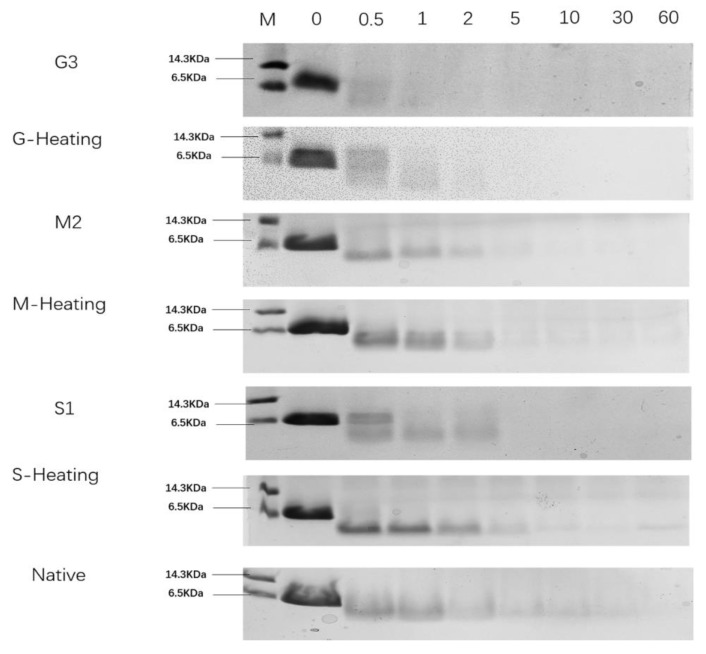
SDS-PAGE of Native and glycated parvalbumin heated by a microwave or an induction cooker in different pepsin digestion time. G3 (Glucose, 1.5 mol/L, 15 min), M2 (Maltose, 1 mol/L, 15 min), and S1 (Sucrose, 0.5 mol/L, 15 min) were heated by a microwave; G-Heating (Glucose, 1.5 mol/L, 15 min), M-Heating (Maltose, 1.5 mol/L, 15 min), and S1 (Sucrose, 1.5 mol/L, 15 min) were heated by an induction cooker. Native was unheated parvalbumin. MW Marker was indicated for lanes labeled ‘‘M.’’.

**Table 1 foods-12-00052-t001:** Orthogonal experiments of the microwave-assisted Maillard reaction.

Experimental Group	Group Name	Factors			Browning Index
		A	B	C	
Orthogonal experiment group ^a^	G1	1	1	1	0.150
	G2	1	2	2	0.220
	G3	1	3	3	0.345 **
	M1	2	1	2	0.105
	M2	2	2	3	0.198 **
	M3	2	3	1	0.070
	S1	3	1	3	0.145 **
	S2	3	2	1	0.056
	S3	3	3	2	0.065
Source of variation	SS	df	MS	F	*p*
Type of sugar (A)	0.037	2	0.018	23.777	<0.05
concentration (B)	0.001	2	0.001	0.832	>0.05
heating time (C)	0.030	2	0.015	19.396	<0.05
Error	0.001	2	0.001		
Total variation	0.075	8			
Control group ^b^	G-Heating	1	3	3	0.062 **
	M-Heating	2	3	3	0.060 **
	S-Heating	3	3	3	0.051 **

Note: G1 (Glucose, 0.5 mol/L, 5 min), G2 (Glucose, 1.0 mol/L, 10 min), G3 (Glucose, 1.5 mol/L, 15 min), M1 (Maltose, 0.5 mol/L, 10 min), M2 (Maltose, 1 mol/L, 15 min), M3 (Maltose, 1.5 mol/L, 5 min), S1 (Sucrose, 0.5 mol/L, 15 min), S2 (Sucrose, 1.0 mol/L, 5 min) and S3 (Sucrose, 1.5 mol/L, 10 min) were heated by a microwave; M-Heating (maltose, 1.5 mol/L, 15 min), S-Heating (sucrose, 1.5 mol/L, 15 min). ^a^ Samples in the orthogonal experiment group were heated by a microwave. ^b^ Samples in the control group were heated by an induction cooker; ** means that there is a significant difference between the samples (*p* < 0.01).

**Table 2 foods-12-00052-t002:** Summary of the theoretical and observed tryptic peptide masses from parvalbumin isoforms PV-Ⅰ and PV-Ⅱ in the samples G3 and M2.

Sample	Protein	Peptide Location	Native Peptide Sequence	Modified Peptide Sequence	Theoretical Mass	Observed Mass
G3	PV-I	34–45	VGLTAKTPDEIK	VGLTAK[G]TPDEIK	1270.71	1432.76
				VGLTAKTPDEIK[Arg]	1270.71	1426.81
G3	PV-I	46–55	KAFEVIDQDK	K[Cmc]AFEVIDQDK	1191.61	1248.61
				K[Frm]AFEVIDQDK	1191.61	1219.61
G3	PV-I	47–55	AFEVIDQDK	AFEVIDQDK[G]	1063.52	1224.58
G3	PV-I	77–88	ALTDAETKVFLK	ALTDAETK[G]VFLK	1334.74	1496.79
				ALTDAETK[Cmc]VFLK	1334.74	1392.74
G3	PV-I	89–108	AGDSDGDGKIGIDEFAVM[O]VK	AGDSDGDGK[G]IGID EFAVM[O]VK	2038.95	2201.00
G3	PV-II	77–88	ALTDAETATFLK	ALTDAETA[Frm]TFLK	1279.67	1307.67
				ALTDA[Frm]ETATFLK	1279.67	1307.67
M2	PV-I	34–45	VGLTAKTPDEIK	VGLTAK[Frm]TPDEIK	1270.71	1298.71
M2	PV-I	77–88	ALTDAETKVFLK	ALTDAETK[Frm]VFLK	1334.74	1362.74
				ALTDAETK[Cmc]VFLK	1334.74	1392.74
				ALTDAETK[*]VFLK	1334.74	1431.77
				ALTDAETKVFLK[Frm]	1334.74	1362.74
				ALTDAETKVFLK[Cmc]	1334.74	1392.74
M2	PV-II	77–88	ALTDAETATFLK	ALTD[Frm]AETATFLK	1279.67	1307.67
				ALT[Frm]DAETATFLK	1279.67	1307.67

## Data Availability

Data are contained within the article.

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
