# Peer review of "Structural Property, Immunoreactivity and Gastric Digestion Characteristics of Glycated Parvalbumin from Mandarin Fish (Siniperca chuaisi) during Microwave-Assisted Maillard Reaction"

_foods, 2022, doi:10.3390/foods12010052_

Round 1

Reviewer 1 Report

Dear authors, the work reveals a high quality research and is very well written, however I leave some considerations:

Point 2.3 - do you always do the repetitions in triplicate as described in 2.9? If not, the number of repetitions should be presented in each point. Still in this topic you do not describe how the results are demonstrated. Are they presented in absorbance? Table 1 presents an index... Can you clarify this situation?

Line 115 - replace hour by h

Line 157 and line 84, write the name of the reagent in the same way

2.8 - how are the results represented?

2.9 - not very descriptive topic. Which test(s) are used?

3. I propose to replace "results" by "results and discussion

Table 1 - is confusing, should explain what the factors are and what the capital letters A, B and C represent, as well as highlighting the identification of the groups (a and b, lowercase, first column)

Figure 2 - looks like it is repeating information from table 2, consider its presence in appendix.

Line 379 - I think it will be important to mention in the conclusion the sugar with the best result. 

Author Response

Response to Reviewer 1 Comments

Point 1: do you always do the repetitions in triplicate as described in 2.9? If not, the number of repetitions should be presented in each point. Still in this topic you do not describe how the results are demonstrated. Are they presented in absorbance? Table 1 presents an index... Can you clarify this situation?

Response 1: Thank you for pointing this out. All experimental repetitions are consistent with those described in 2.9. Index in Tabel 1 refers to the browning index of Maillard reaction, which is the absorbance value obtained at a wavelength of 420 nm, which has been clarified in Table 1.

Point 2: Line 115 - replace hour by h

Response 2: Thank you for pointing this out.It has been revised in Line 126.

Point 3:Line 157 and line 84, write the name of the reagent in the same way

Response 3: Thank you for pointing this out. It has been revised in Line 172-173.

Point 4: 2.8 - how are the results represented?

Response4: Thank you for your comment. Resistance of protein to pepsin digestion can be used as a marker of potential al-lergenicity. The digestibility of glycated parvalbumin samples from different heating methods was verified by SDS-PAGE and the specific results are shown in Figure 5.

Point5: 2.9 - not very descriptive topic. Which test(s) are used?

Response5: Thank you for pointing this out. It has been revised in Line 196-198.

Point5: 6. I propose to replace "results" by "results and discussion

Response 6: Thank you for pointing this out. It has been revised in the manuscript Line 199.

Point 7 :Table 1 - is confusing, should explain what the factors are and what the capital letters A, B and C represent, as well as highlighting the identification of the groups (a and b, lowercase, first column)

Response 7: Thank you for pointing this out. It has been revised in the manuscript Line214-221.

Point 8: Figure 2 - looks like it is repeating information from table 2, consider its presence in appendix.

Response8: Thank you for your comment. Figure 2 (a) shows the details of peptide detected by mass spectrometry during protein identification, and Figure 2 (b) shows the fragment attribution of the MS/MS spectrum of the identified peptide.It is not a simple repetition of Table 2.

Point 9: Line 379 - I think it will be important to mention in the conclusion the sugar with the best result. 

Response 9: Thank you for pointing this out. It has been revised in the manuscript Line411-412.

Reviewer 2 Report

Reviewer comments:

In the manuscript (foods-2024062) the authors have investigated the effect of parvalbumin`s gycation during microwave-assisted Maillard reaction on its structure, immunogenicity and susceptibility to digestion by pepsin.  

The manuscript can be very interesting for readers, but it needs significant improvement in the sections Material and methods, and Results. In this form, the manuscript needs major corrections, Please, take into account below some of the comments and suggestions for the improvement of your article's quality, before the final decision.

In the Material and methods section, some parts of the experiment are mixed and under different titles, so it is not clear which conditions were in the specific experiment (for example line 90, 98-102, and 114-117). For example it is better to insert the new titles such as: Orthogonal designed experiment, Preparation of incubation mixtures, Preparation of crude extract and so on.

All experiments should be described in more detail. In all experiments some key data are missing such as;

The final concentrations of parvalbumin in the reaction mixtures and crude extract, as well as the starting concentration of parvalbumin and the degree of antigen dilution in indirect ELISA are missing. It is not clear, is a crude extract prepared before or after heating experiments? Which concentration of parvalbumin was used for the production of antibodies, and how were they isolated? Description of the heating experiment using induction cooker and conditions in the experiment should be added.

Authors should explain why they additionaly reduce the number of mixtures in the control group, only 3 mixtures (induction cooker) compared to 9 mixtures used during microwave heating. For the statistical analysis (Table 1) you need to have an appropriate control group.  They also need to explain the criteria for the selection of only 7 samples in the experiments where parvalbumin immunological properties and digestibility were investigated.

In the results section there are discrepancies between explanation of the results in the text and results shown in Tables and figures. These are some examples: lines 224-227, it is not clear how the value 66.03% was obtained or calculated for the inhibition effect G3, when the maximal inhibition for native parvalbumin is 55% according to the figure 4. The explanation in lines 341 and 342 is in discrepancy with figure 5. Please carefully check the whole results section. The sentences as well as order of the text in the results section need to be improved. 

This is only one example of sentence how the authors can improve their manuscript. Sucrose is not a reducing sugar with a ketone group, which are not active in itself and can work only through degradation steps, can be replace with the sentence:  Sucrose is not a reducing sugar with a free carbonyl group and the activation of this group can be achieved  only through degradation steps of sugar, or  Sucrose belong to the non-reducing sugars in which carbonyl groups of two sugars are  involved in the formation of glycosidic bond, so these sugars can be accessible for the Maillard reaction with proteins only after degradation steps.

Tables and Figures as well as their titles and legends should be corrected. For example Table 1, superscript a and b should be A and B, while c, d and e should be replace with *, **, #, or something like that. In legend of figure 1, there are repetitions…The order of peptide sequence should be in one row, and order of MS spectra in SM should be the same as order of peptides in Table 2..please check other Tables and figures in the manuscript.

Beside these major corrections, also minor technical corrections such as : use of abbreviations in the text, text formatting, the style of references cited  (for example Kue et al replace with Kuehn et al (2017), line 296 NCBI …). It is better to use reaction mixture instead of group. Do not start the sentence with the number like in line 155, and so on... 

Author Response

Response to Reviewer 2 Comments

Point 1: In the Material and methods section, some parts of the experiment are mixed and under different titles, so it is not clear which conditions were in the specific experiment (for example line 90, 98-102, and 114-117). For example it is better to insert the new titles such as: Orthogonal designed experiment, Preparation of incubation mixtures, Preparation of crude extract and so on.

Response 1: Thank you for pointing this out. New titles have been added to Materials and Methods section to explain experimental details (Line 90- 120).

Point 2: All experiments should be described in more detail. In all experiments some key data are missing such as;The final concentrations of parvalbumin in the reaction mixtures and crude extract, as well as the starting concentration of parvalbumin and the degree of antigen dilution in indirect ELISA are missing.

Response 2: Thank you for pointing this out. The crude extract is a mixture, the concentration of crude extracted protein in the manuscript is 2mg/mL, and the details have been added in the manuscript Line 117- 120. The concentration of parvalbumin as antigen is 2 μg/mL, this has been explained in the manuscript Line 163.

Point 3: It is not clear, is a crude extract prepared before or after heating experiments?

Response 3: Thank you for pointing this out, the crude extract prepared was after heating experiments,this has been explained in the manuscript Line 114.

Point 4: Which concentration of parvalbumin was used for the production of antibodies and how were they isolated?

Response 4: Thank you for pointing this out, parvalbumin was purified with 60% saturated sulfate, and the details have been added in the manuscript Line 127- 130. The concentration of parvalbumin used to produce antibodies was 2mg/mL and the details have been added in the manuscript Line 159-162.

Point 5: Description of the heating experiment using induction cooker and conditions in the experiment should be added.

Response 5: Thank you for pointing this out. In the heating experiment, the use conditions of the cooker  have been supplemented in Line 111-112.

Point 6: Authors should explain why they additionaly reduce the number of mixtures in the control group, only 3 mixtures (induction cooker) compared to 9 mixtures used during microwave heating. For the statistical analysis (Table 1) you need to have an appropriate control group.  They also need to explain the criteria for the selection of only 7 samples in the experiments where parvalbumin immunological properties and digestibility were investigated.

Response 6: Thank you for your comment. According to the previous literature, heating method, sugar type, sugar concentration and heating time are the four main factors affecting the browning index of Maillard, among which the influence of induction cooker on the browning index of Maillard is smaller than that of microwave-assisted heating, and thesingle-factor also prove this. Under the condition that the heating method is an induction cooker, the highest browning index of Maillard was selected from three different sugar treatment groups, namely G-Heating (glucose, 1.5 mol/L, 15 min), M-Heating (maltose, 1.5 mol/L, 15 min), S-Heating (sucrose, 1.5 mol/L, 15 min). Taking G-Heating, M-Heating and S-Heating as controls, this can not only objectively evaluate the browning index of induction cooker, but also illustrate the advantages and effects of microwave-assisted heating compared with induction cooker heating from the experimental results.

Point 7: They also need to explain the criteria for the selection of only 7 samples in the experiments where parvalbumin immunological properties and digestibility were investigated.

Response7: Thank you for your comment. Only 7 samples were selected for test immunological properties and digestibility of parvalbumin. The reason for this was that according to the orthogonal experimental results and SDS-PAGE analysis, it was proved that the glycosylation of G3 (glucose, 1.5 mol/L, 15 min) was the best in the glucose treatment group. The glycosylation effect of M2 (maltose, 1 mol/L, 15 min) was the best in the maltose treatment group, and the glycosylation of S1 (sucrose, 0.5 mol/L, 15 min) was the best in the sucrose treatment group, which is also in line with the purpose of orthogonal experimental screening of optimal microwave treatment conditions, and provides a theoretical basis for desensitization in daily life.

8.In the results section there are discrepancies between explanation of the results in the text and results shown in Tables and figures. These are some examples: lines 224-227, it is not clear how the value 66.03% was obtained or calculated for the inhibition effect G3, when the maximal inhibition for native parvalbumin is 55% according to the figure 4. The explanation in lines 341 and 342 is in discrepancy with figure 5. Please carefully check the whole results section. The sentences as well as order of the text in the results section need to be improved.

Response 8: Thank you for your comment. Those two value obtained compared with the inhibition rate of untreated group. The specific calculation method was as follows: (inhibition rate of untreated group - inhibition rate of treated group)/inhibition rate of untreated group.

89.This is only one example of sentence how the authors can improve their manuscript. Sucrose is not a reducing sugar with a ketone group, which are not active in itself and can work only through degradation steps, can be replace with the sentence:  Sucrose is not a reducing sugar with a free carbonyl group and the activation of this group can be achieved  only through degradation steps of sugar, or  Sucrose belong to the non-reducing sugars in which carbonyl groups of two sugars are  involved in the formation of glycosidic bond, so these sugars can be accessible for the Maillard reaction with proteins only after degradation steps.

Response9: Thank you for your comment. The whole manuscript has been grammatically/editorially reviewed and the sentence has been revised as “Sucrose is not a reducing sugar…” in the manuscript Line 211-212.

10.Tables and Figures as well as their titles and legends should be corrected. For example Table 1, superscript a and b should be A and B, while c, d and e should be replace with *, **, #, or something like that. In legend of figure 1, there are repetitions…The order of peptide sequence should be in one row, and order of MS spectra in SM should be the same as order of peptides in Table 2..please check other Tables and figures in the manuscript.

Response10: Thank you for pointing this out. Since the letter A, B has been annotated as different factors in Table 1, the annotation method of a,b superscript has been retained in Line 212-213.The letter note (c,d,e) have been revised in the manuscript and the peptide sequence in Tabel 2 has been adjusted. The order of peptide sequence in SM also has been revised.

11.Beside these major corrections, also minor technical corrections such as : use of abbreviations in the text, text formatting, the style of references cited  (for example Kue et al replace with Kuehn et al (2017), line 296 NCBI …). It is better to use reaction mixture instead of group. Do not start the sentence with the number like in line 155, and so on... 

Response11: Thank you for pointing this out. The sentence has been revised in the manuscript Line 46, Line 50 and Line 63. The mistake of starting a sentence with a number has been revised and adjusted in whole manuscript.

Reviewer 3 Report

Dear Authors,

Below, I am sending my opinion of manuscript entitled: “Structural Property, Immunoreactivity and In Vitro Digestion Characteristics of Glycated Parvalbumin from Mandarin Fish (Siniperca chuaisi) during Microwave-Assisted Maillard Reaction” number 2024062.

The work is very interesting and in my opinion presented issue in good style has been described. See attached pdf file, I am sending minor revision of the paper.

The Title: no changes, adequate to contents.

The Abstract is written in good style and I think all needed information were placed in this section.

Keywords: corresponds to the content

The Material and Methods: described with all details, small correction needed.

Results and Discussion: the authors very clear described the changes taking place during thermal processing (microwave and induction cooker) depending on type and concentration of used sugars. The problem has been solved using proper analytical methods such as: electrophoresis and HPLC-MS. Therefore the obtained observations and results have been confirming by using other techniques. Finally the authors recommended thermal processing or daily food cooking in the aim to decrease allergenicity of parvalbumin from mandarin fish.

Final decision: in my opinion the paper was written in good style

Small comments also in attached pdf document.

Author Response

Response to Reviewer 3 Comments

Point 1: . Please standarise: if you use cat number, please place in all places, or delete from pepsin.

Response 1: Thank you for pointing this out.The cat number of Pepsin has been removed.

Point 2: SDS-PAGE, LC-MS-MS, ACN, FA, ELISA and many other abbreviations must be defined.

Response 2: Thank you for pointing this out. The full name of the abbreviation has been added to the manuscript.

Point 3: “ Results ”should be“Results and Discussion ”

Response 3: Thank you for pointing this out.The point has been revised .

Point 4:  “traditi.onal”

Response 4: Thank you for pointing this out. It has been revised in Line 216.

Point 5: Remove the sentence(The structural and immunological properties of glycated mandarin fish parvalbumin from microwave-assisted Maillard reaction were…) in conclusion section .

Response 5: Thanks for your comment. As a conclusion, it serves as a summary of this study, and this sentence has been revised in the manuscript.

Point 6:The list of references needs to be shortened

Response 6: Thanks for your comment. The literature is closely related to the research of manuscripts, so all references are retained.

Reviewer 4 Report

The work presents interesting results, but I have some comments for the authors.

In the title and in the abstract it should be indicated that the digestion test was the gastric phase. In vitro gastric digestion

All used abbreviations must be defined when first used. There are abbreviations that are very familiar to science readers, but it must be considered that the work may have readers from different areas. SDS-PAGE, LC-MS-MS, ACN, FA, ELISA and many other abbreviations must be defined.

The way to include the bibliographical references in the manuscript must comply with the journal's standards [1], [2,3], [4-6]. Please correct throughout the manuscript.

Line 92 : sugar concentration (0.5 mol/L, 1 mol/L, 1.5 mol/L). Change for sugar concentration (0.5;1.0 and 1.5 mol/L)

Line 93; time(5min, 10min, 15min). Change for time (5, 10, and, 15 min). Correct this type of errors in the manuscript please.

Line 115: After standing for 2 hours, Change for After standing for 2 h… The format must be homogeneous, sometimes hours are used and others h. The comment applies to other terms used in the manuscript.

Table 1. In requesting the table, all the terms used must be defined. The table must be interpreted alone without the need to refer to the text of the manuscript. Define G1, G2, G3, M1, M2…

The bibliographical references section must be corrected according to the rules for authors. For example, the journal name should be abbreviated. « Ávila-Román, J.; Soliz-Rueda, J.R.; Bravo, F.I.; Aragones, G.; Suarez, M.; Arola-Arnal, A.; Mulero, M.; Salvadó, M.J.; Arola, L.; Torres-Fuentes, C.; et al. Phenolic Compounds and Biological Rhythms: Who Takes the Lead? Trends Food Sci. Technol. 2021, 113, 77–85”.

Author Response

Point 1: . In the title and in the abstract it should be indicated that the digestion test was the gastric phase. In vitro gastric digestion.

Response 1: Thank you for pointing this out. revised. This point has been revised .

Point 2: All used abbreviations must be defined when first used. There are abbreviations that are very familiar to science readers, but it must be considered that the work may have readers from different areas. SDS-PAGE, LC-MS-MS, ACN, FA, ELISA and many other abbreviations must be defined.

Response 2: Thank you for pointing this out. The full name of the abbreviation has been added to the manuscript.

Point 3: The way to include the bibliographical references in the manuscript must comply with the journal's standards [1], [2,3], [4-6]. Please correct throughout the manuscript.

Response 3: Thank you for pointing this out.The application format of the reference has been revised.

Point 4: Line 92 : sugar concentration (0.5 mol/L, 1 mol/L, 1.5 mol/L). Change for sugar concentration (0.5;1.0 and 1.5 mol/L)ï¼›Line 93; time(5min, 10min, 15min). Change for time (5, 10, and, 15 min). Correct this type of errors in the manuscript please.Line 115: After standing for 2 hours, Change for After standing for 2 h… The format must be homogeneous, sometimes hours are used and others h. The comment applies to other terms used in the manuscript.

Response 4: Thank you for pointing this out. It has been revised in Line 100, Line 127.

Point 5: Table 1. In requesting the table, all the terms used must be defined. The table must be interpreted alone without the need to refer to the text of the manuscript. Define G1, G2, G3, M1, M2…

Response 5: Thank you for pointing this out. It has been defined in Line 215-218.

Point 6: The bibliographical references section must be corrected according to the rules for authors. For example, the journal name should be abbreviated. « Ávila-Román, J.; Soliz-Rueda, J.R.; Bravo, F.I.; Aragones, G.; Suarez, M.; Arola-Arnal, A.; Mulero, M.; Salvadó, M.J.; Arola, L.; Torres-Fuentes, C.; et al. Phenolic Compounds and Biological Rhythms: Who Takes the Lead? Trends Food Sci. Technol. 2021, 113, 77–85”

Response 6: Thank you for pointing this out. The references in the manuscript has been revised.

Round 2

Reviewer 2 Report

Reviewer comments

I need to say that the authors did not make big efforts to correct all mistakes in the first version of the manuscript. They did not even correct visible technical mistakes, which makes additional work for reviewers. Please, take into account all suggestions listed below to improve your manuscript before the final decision. In the manuscript all need corrections are marked.

Introduction

Line 39 and 40

Please, insert species after fish.

Line 40, 60

Insert year of publication after Yang et al.  Insert in the manuscript point after et al.

Line 46

Insert fish after mandarin.

Line 48

Space before there.

Line 59

Replace protein was with proteins were.

Line 60

Replace its sensitization with their allergenicity.

Line 70

Formatting in mandarin.

Lines 72, 73

Replace sentence:  The effect of structural, immunological properties, as well as the digestibility of  glycated parvalbumin with Maillard reaction were carefully studied., with

The main goal of this study was investigation how Maillard reaction affects the structural and immunological properties of parvalbumin and its gastric digestibility.

Lines 77,78

Replace sentence with:

The relationship between structural modification of parvalbumin by Maillard reaction and its immunoreactivity was also investigated using indirect competitive ELISA assay.

Material and methods

Line 85

Insert (FA) and (ACN) after Formic acid and acetonitrile

Lines 90-95

Formatting

Lines 93-95

Replace sentences with:

Then, the fish flesh was washed three times with distilled water and smashed using an ultramicro pulverizer. Finally, 20 g of fish flesh were accurately weighed, packed and stored at -80 °C until experiments.

Lines 97-109

Replace with:

To reduce the number of experiments the orthogonal designed experiment approach was applied. According to the sugar commonly used in household and food industry production, glucose, maltose and sucrose were selected as a first factor (A). Beside the type of sugar, according to the results of the pre-experiment, sugar concentration (0.5, 1, 1.5 mol/L) (B) and microwave heating time (5, 10, 15min) (C) were selected as additional two factors in the orthogonal designed experiments. Nine experimental conditions were designed: G1 (glucose, 0.5 mol/L, 5 min), G2 (glucose, 1 mol/L, 10 min), G3 (glucose, 1.5 mol/L, 15 min), M1 (maltose, 0.5 mol/L, 10 min), M2 (maltose, 1 mol/L, 15 min), M3 (maltose, 1.5 mol/L, 5 min), S1 (sucrose, 0.5 mol/L, 15 min), S2 (sucrose, 1 mol/L, 5 min), and S3 (sucrose, 1.5 mol/L, 10 min). Another three experimental conditions were selected as a control: G-Heating (glucose, 1.5 mol/L, 15 min), M-Heating (maltose, 1.5 mol/L, 15 min), S-Heating (sucrose, 1.5 mol/L, 15 min). The high flame heating (2450 MHz) was selected as a condition during microwave-assisted Maillard reaction, while 800 W was selected during induction cooker-assisted Maillard reaction (control). In all experiments, the material-liquid ratio (w/v) was 1:4.

Line 112

It should be specified here do you take only part (20g) of reaction mixture (for the heating experiments 20g of fish flesh was mixed with 80 mL of sugar solution), or you take whole mixture.

Line 112

Insert rigorously before homogenized

Line 114, 115

Then, the homogenate was centrifuged 10 min at 10000g for 10 min at 4 ℃ with, and supernatant was collected as a crude extract. The protein concentration in crude extract was determined by Bradford method using bovine serum albumin as a standard.

Line 118

Replace sample with proteins in the crude extracts  …by dilution with (please insert did you used buffer or water for the dilution) .

Lines 120-122

Browning index as an indicator of the degree of Maillard reaction can be estimated by the measurement of absorbance at 420 nm. Absorbance at 420 nm in the diluted samples of crude extracts (200 µL) was measured using a microplate plate reader (please insert Type of device and company).

Line 124

space

Line 125

Insert diluted before crude

Line 126

Please insert in which ratio these two solution were mixed, in other word which was the final saturation of ammonium sulfate.

Replace r/min with rpm/min

Line 127, 128

The supernatant as fish extract was collected. The supernatant as fish extract was 127 collected and stored at -20 °C until analysis.

Please, you need to exactly specified is parvalbumin in the supernatant fraction or in the precipitate. So if it is in the supernatant then the sentence should be: The supernatant, a partially purified paralbumin fraction, was collected and stored at -20 °C until analysis.

Line 130

Replace fish extract with purified parvalbumin , boiled in with mixed with 4x…

Line 131

Insert …of proteins 1mg/mL for 5 min 95℃, Electrophoresis was conducted at a constant … and gels were …

Line 133

 weight of parvalbumin in the samples was

Line 136

Did you perform in gel digestion or in solution digestion?

Line 141

Abbreviations  are given in line 85, delete full names

Line 157, 158

Replace PV with parvalbumin

Line 154

Please replace existing title with Immunization of rabbit by mandarin fish parvalbumin and preparation of antiserum

Specific conditions are as follows: 2 mg/mL purified PV (was mixed with Freund's incomplete adjuvant 1:1, and New Zealand white rabbits were injected intramuscularly at the weight of.

Animals were immunized by intramuscular injection of parvalbumin in concentration of 0.5 mg/kg. Parvalbumin solution for immunization was prepared by mixing in ratio 1:1 purified   parvalbumin solution (2.2 mg/mL, purity higher than 85%) and Freund's incomplete adjuvant.

Did animal receive one or several doses? How many animals did you use for immunization?

Lines 162-167

Description of experiment should be in next order:

The coating of microplate with antigen (parvalbumin) during overnight,

Blocking with milk proteins,

Biding of antibodies from prepared mixtures

Description how mixtures of rabbit antiserum and parvalbumin from different samples were prepare.

Washing

Biding of the second antibody

Washing

Enzyme reaction

Please correct this chapter, also add rabbit before antiserum.

Line 182

Gastric, add purified parvalbumin samples after heating experiment

Line 185

In the line 117, you stated that final concentration of parvalbumin was adjust 2 mg/mL. Please add this concentration here or correct line 185 was adjusted to 2 or 5 mg/ml

Lines 186-187

These sentences should be corrected like …placed in water bath to incubate during 60 min at 37℃. At time points 0,5, 1, …. initiation of incubation, 200 µL of digestion mixture was removed and mixing with 70 µL…. to stop enzyme reaction.  After that samples were heated 10 min at 95℃ and analyzed SDS-PAGE.

Results and discussion

Line 203

Replace deeper with higher

Line219-220

Please delete, it is two times repeat.

Line 203

Insert after Table1, according to the type of sugar tested in the orthogonal experiment the highest….were obtained for G3….

Please delete the conditions for

Line 210

Replace marked with reducing sugars with free carbonyl group,

Line 213

Replace groups with two sugars

Lines 224 225

Replace marked part with

the values of browning index obtained during microwave heating was significantly higher compared to the values obtained using induction cooker, when compared G3/G-Heating,  M2/M-Heating and S1/S-Heating.

Line 227

Replace with yield of

Line 228, 229

Replace groups with samples

Line 233

Insert that before sugar

Line235,  236

Replace with: The main reason is the different types of sugar and raw material states used in the Maillard reaction as well as conditions of heating.

Line 241

More than one authors so it is better Yang`s replace with previous

Line 243  replace pattern with analysis

Line 244 replace proteins with parvalbumin

Line 246

Replace a with b

Line 247 and 248

Replace with

the glycated parvalbumin from M2 (maltose, 1 mol/L, 15 min) gave the shortest molecular weight migration, followed by G3 (glucose, 1.5 mol/L, 15 min)

Line 252  insert Figure 1a after 10 min) ; Replace Teodorowicz`s with Teodorowicz et al (2013)

Line 261 Insert compare to G1 and G2 between SDS-PAGE and indicating

Line 264 and 265 Delete marked part, , line 266 insert Figure 1b after parvalbumin

Line 267

Replace marked part with The similar pattern was obtain for samples M2 and M-Heating

Replace level with yield

Line 274 Replace with samples obtained during

Line 283- 288

Sentences replace with

To investigate the glycation of parvalbumin, samples G3 and M2 with best yield of Maillard reaction were chosen to be analyzed by Nano-LC-MS/MS. Native mandarin fish parvalbumin was also analyzed as control. Amino acid analysis of tryptic peptides obtained from native parvalbumin and Uniprot database matching, showed two isoforms of parvalbumin, named PV-I (99% coverage) and PV-II (75% coverage), respectively (data shown as Figure 2).

After this insert sentences from lines 294-298 PV-I and PV-II are the most common parvalbumin isoforms, their protein structure are similar, both of them contain 108~109 amino acids, but the amino acid sequence is different, and the homology of sequence is only 50 to 80%. Both isoforms can cause allergic reactions, but PV-I has stronger binding ability with IgG and IgE than PV-II[38]

Lines 289- 293

Replace products from with samples in line 289. This part move after sentences in lines 294-297

Replace N with n

Line 317, 318

Insert parvalbumin isoforms before PV-I, replace products from with samples.

Line 332  A replace with a

Line 326 add et al. and year

Line 329

Replace marked text with Marchler-Baure et a. (2015)[Number of reference]

Line 336

Immunogenicity

Lines 343

Replace with Amino acid sequence of parvalbumin isoforms PV-I (a), and PV-II (b) in mandarin fish.

Lines 339-341

Replace with Peptide matching diagram of parvalbumin isoforms PV-I (a), and PV-II (c) in native sample and their mass spectrograms (b) and (d), respectively.

Lines 352

Replace marked part  with all samples obtained during both type of heating (microwave or induction cooker) were lower

When compare the inhibition rates of different sugars (samples G3, M2 and S1) to the inhibition rate of native parvalbumin, it was found that glucose had the highest inhibition effect, up to 66.03%,  followed maltose and sucrose,

Line 357-359

Replace with It was found that the inhibition rates of samples obtained during microwave heating were significantly lower than those obtained during induction cooker heating.

Replace There with These

Lines 375-377

The conclusion about digestibility G3 and G-Heating samples cannot clearly see from the figure. After 2 minutes there is no visible protein bands in both samples.

Refrences

Because reference 21 is not cited in the text,  and order of some part of text was changed it should change reference number  in the manuscript.

Author Response

Response to Reviewer 2 Comments

Dear  Reviewer,

Thanks very much for taking your time to review this manuscript. We really appreciate all your generous comments and suggestions! Please find my itemized responses in below and my revisions in the re-submitted files.

Thanks for your kindness and best wishes.

Yours sincerely,

Dr. Hong Zhang

Point 1: Line 39 and 40  Please, insert species after fish.

Response 1: Thank you for pointing this out. It has been revised in the manuscript Line 39-40.

Point 2: Line 40, 60 Insert year of publication after Yang et al.  Insert in the manuscript point after et al.

Response 2: Thank you for pointing this out. It has been revised in the manuscript Line 49, 61.

Point 3: Line 46 Insert fish after mandarin.

Response 3: Thank you for pointing this out. It has been revised in the manuscript Line 47.

Point 4: Line 48  Space before there.

Response 4: Thank you for pointing this out. It has been revised in the manuscript Line 49.

Point 5: Line 59 Replace protein was with proteins were.

Response 5: Thank you for pointing this out. It has been revised in the manuscript Line 60.

Point 6: Line 60  Replace its sensitization with their allergenicity.

Response 6: Thank you for your comment. It has been revised in the manuscript Line 61.

Point 7: Line 70  Formatting in mandarin.

Response7: Thank you for your comment. It has been revised in the manuscript Line 71.

  1. Lines 72, 73 Replace sentence: The effect of structural, immunological properties, as well as the digestibility of glycated parvalbumin with Maillard reaction were carefully studied., with

The main goal of this study was investigation how Maillard reaction affects the structural and immunological properties of parvalbumin and its gastric digestibility.

Response 8: Thank you for your comment. It has been revised in the manuscript Line 73-74.

  1. Lines 77,78 Replace sentence with:The relationship between structural modification of parvalbumin by Maillard reaction and its immunoreactivity was also investigated using indirect competitive ELISA assay.

Response9: Thank you for your comment. It has been revised in the manuscript Line 78-79.

  1. Line 85 Insert (FA) and (ACN) after Formic acid and acetonitrile.

Response10: Thank you for pointing this out. It has been revised in the manuscript Line 87.

  1. Lines 90-95 Formatting

Response11: Thank you for pointing this out.

  1. Lines 93-95 Replace sentences with:Then, the fish flesh was washed three times with distilled water and smashed using an ultramicro pulverizer. Finally, 20 g of fish flesh were accurately weighed, packed and stored at -80 °C until experiments.

Response12: Thank you for pointing this out. It has been revised in the manuscript Line 96-99.

  1. Lines 97-109 Replace with:To reduce the number of experiments the orthogonal designed experiment approach was applied. According to the sugar commonly used in household and food industry production, glucose, maltose and sucrose were selected as a first factor (A). Beside the type of sugar, according to the results of the pre-experiment, sugar concentration (0.5, 1, 1.5 mol/L) (B) and microwave heating time (5, 10, 15min) (C) were selected as additional two factors in the orthogonal designed experiments. Nine experimental conditions were designed: G1 (glucose, 0.5 mol/L, 5 min), G2 (glucose, 1 mol/L, 10 min), G3 (glucose, 1.5 mol/L, 15 min), M1 (maltose, 0.5 mol/L, 10 min), M2 (maltose, 1 mol/L, 15 min), M3 (maltose, 1.5 mol/L, 5 min), S1 (sucrose, 0.5 mol/L, 15 min), S2 (sucrose, 1 mol/L, 5 min), and S3 (sucrose, 1.5 mol/L, 10 min). Another three experimental conditions were selected as a control: G-Heating (glucose, 1.5 mol/L, 15 min), M-Heating (maltose, 1.5 mol/L, 15 min), S-Heating (sucrose, 1.5 mol/L, 15 min). The high flame heating (2450 MHz) was selected as a condition during microwave-assisted Maillard reaction, while 800 W was selected during induction cooker-assisted Maillard reaction (control). In all experiments, the material-liquid ratio (w/v) was 1:4.

Response 13: Thank you for your comment. It has been revised in the manuscript Line 100-114.

  1. Line 112 It should be specified here do you take only part (20g) of reaction mixture (for the heating experiments 20g of fish flesh was mixed with 80 mL of sugar solution), or you take whole mixture.

Response14: Thank you for your comment. It has been revised in the manuscript Line 116.

  1. Line 112 Insert rigorously before homogenized

Response15: Thank you for pointing this out. It has been revised in the manuscript Line 116.

  1. Line 114, 115 Then, the homogenate was centrifuged 10 min at 10000g for 10 min at 4 ℃ with , and supernatant was collected as a crude extract. The protein concentration in crude extract was determined by Bradford method using bovine serum albumin as a standard.

Response16: Thank you for pointing this out. It has been revised in the manuscript Line 118-120.

  1. Line 118 Replace sample with proteins in the crude extracts …by dilution with (please insert did you used buffer or water for the dilution) .

Response17: Thank you for pointing this out. It has been revised in the manuscript Line 121.

  1. Lines 120-122 Browning index as an indicator of the degree of Maillard reaction can be estimated by the measurement of absorbance at 420 nm. Absorbance at 420 nm in the diluted samples of crude extracts (200 µL) was measured using a microplate plate reader (please insert Type of device and company).

Response18: Thank you for pointing this out. It has been revised in the manuscript Line 121-127.

  1. Line 124 space

Response19: Thank you for pointing this out. It has been revised in the manuscript.

20.Line 125 Insert diluted before crude

Response20: Thank you for pointing this out. It has been revised in the manuscript Line129.

  1. Line 126 Please insert in which ratio these two solution were mixed, in other word which was the final saturation of ammonium sulfate.Replace r/min with rpm/min

Response21: Thank you for your comment. This is a method to purify protein with saturated sulfate. The volume of saturated sulfate added is calculated according to the volume of crude protein extract. For example, add 750 mL saturated sulfate concentration (100%) to 500 mL protein crude extract to adjust the saturated sulfate concentration to 60%. Therefore, this expression in the manuscript is more accurate and the rpm/min has been revised in the manuscript Line130-132.

  1. Line 127, 128 The supernatant as fish extract was collected.Please, you need to exactly specified is parvalbumin in the supernatant fraction or in the precipitate. So if it is in the supernatant then the sentence should be: The supernatant, a partially purified paralbumin fraction, was collected and stored at -20 °C until analysis.

Response22: Thank you for pointing this out. It has been revised in the manuscript Line 121-127.

  1. Line 130 Replace fish extract with purified parvalbumin , boiled in with mixed with 4x…and Line 131 Insert …of proteins 1mg/mL for 5 min 95℃, Electrophoresis was conducted at a constant … and gels were …

Response23: Thank you for pointing this out. It has been revised in the manuscript Line 134-137.

  1. Line 133 weight of parvalbumin in the samples was

Response24: Thank you for pointing this out. It has been revised in the manuscript Line137.

  1. Line 136 Did you perform in gel digestion or in solution digestion?

Response25: Thank you for pointing this out. It has been revised in the manuscript Line140.

26 Line 141 Abbreviations are given in line 85, delete full names Line 157, 158 Replace PV with parvalbumin

Response26: Thank you for your comment. It has been revised in the manuscript Line 145 and Line 161.

  1. Line 112 It should be specified here do you take only part (20g) of reaction mixture (for the heating experiments 20g of fish flesh was mixed with 80 mL of sugar solution), or you take whole mixture.

Response27: Thank you for your comment. It has been revised in the manuscript Line 116.

  1. Line 154 Please replace existing title with Immunization of rabbit by mandarin fish parvalbumin and preparation of antiserum Specific conditions are as follows: 2 mg/mL purified PV (was mixed with Freund's incomplete adjuvant 1:1, and New Zealand white rabbits were injected intramuscularly at the weight of.Animals were immunized by intramuscular injection of parvalbumin in concentration of 0.5 mg/kg. Parvalbumin solution for immunization was prepared by mixing in ratio 1:1 purified parvalbumin solution (2.2 mg/mL, purity higher than 85%) and Freund's incomplete adjuvant.Did animal receive one or several doses? How many animals did you use for immunization?

Response28: Thank you for pointing this out. It has been revised in the manuscript Line 116.

  1. Lines 162-167 Description of experiment should be in next order:The coating of microplate with antigen (parvalbumin) during overnight,Blocking with milk proteins,Biding of antibodies from prepared mixtures Description how mixtures of rabbit antiserum and parvalbumin from different samples were prepare.Washing.Biding of the second antibody.Washing.Enzyme reaction.Please correct this chapter, also add rabbit before antiserum.

Response 29: Thank you for pointing this out. It has been revised in the manuscript Line 166-175.

30 Line 185 In the line 117, you stated that final concentration of parvalbumin was adjust 2 mg/mL. Please add this concentration here or correct line 185 was adjusted to 2 or 5 mg/ml

Response30: Thank you for pointing this out. It has been revised in the manuscript Line 187.

  1. Lines 186-187 These sentences should be corrected like …placed in water bath to incubate during 60 min at 37℃. At time points 0,5, 1, …. initiation of incubation, 200 µL of digestion mixture was removed and mixing with 70 µL…. to stop enzyme reaction. After that samples were heated 10 min at 95℃ and analyzed SDS-PAGE.

Response 31: Thank you for pointing this out. It has been revised in the manuscript Line 188-192.

  1. Line 203 Replace deeper with higher

Response 32: Thank you for pointing this out. It has been revised in the manuscript Line 203.

  1. Line219-220 Please delete, it is two times repeat.

Response33: Thank you for pointing this out. It has been revised in the manuscript 219-220.

  1. Line 203 Insert after Table1, according to the type of sugar tested in the orthogonal experiment the highest….were obtained for G3….Please delete the conditions for

Response35: Thank you for pointing this out. It has been revised in the manuscript 203-204.

36.Line 210 Replace marked with reducing sugars with free carbonyl group and Line 213 Replace groups with two sugars

Response36: Thank you for pointing this out. It has been revised in the manuscript Line 210 and Line 214 .

37.Lines 224 225 Replace marked part with the values of browning index obtained during microwave heating was significantly higher compared to the values obtained using induction cooker, when compared G3/G-Heating,  M2/M-Heating and S1/S-Heating.

Response37: Thank you for pointing this out. It has been revised in the manuscript Line 224 -226 .

38.Line 227 Replace with yield of and Line 228, 229 Replace groups with samples,Line 233 Insert that before sugar.

Response38: Thank you for pointing this out. It has been revised in the manuscript .

39 Replace with: The main reason is the different types of sugar and raw material states used in the Maillard reaction as well as conditions of heating.Line 241More than one authors so it is better Yang`s replace with previous.Line 243 replace pattern with analysisLine 244 replace proteins with parvalbumin.Line 246 Replace a with b

Response39: Thank you for pointing this out. It has been revised in the manuscript .

40 Line 247 and 248 Replace with the glycated parvalbumin from M2 (maltose, 1 mol/L, 15 min) gave the shortest molecular weight migration, followed by G3 (glucose, 1.5 mol/L, 15 min).

Response40: Thank you for pointing this out. It has been revised in the manuscript Line 247 -248 .

41 Line 252  insert Figure 1a after 10 min) ; Replace Teodorowicz`s with Teodorowicz et al (2013)

Response41: Thank you for pointing this out. It has been revised in the manuscript Line 252 .

42 Line 261 Insert compare to G1 and G2 between SDS-PAGE and indicating

Response42: Thank you for pointing this out. It has been revised in the manuscript Line 261-262 .

43 Line 264 and 265 Delete marked part, , line 266 insert Figure 1b after parvalbumin

Response43: Thank you for pointing this out. It has been revised in the manuscript Line 266.

44 Line 267 Replace marked part with The similar pattern was obtain for samples M2 and M-Heating

Replace level with yield.Line 274 Replace with samples obtained during.Line 283- 288Sentences replace with To investigate the glycation of parvalbumin, samples G3 and M2 with best yield of Maillard reaction were chosen to be analyzed by Nano-LC-MS/MS. Native mandarin fish parvalbumin was also analyzed as control. Amino acid analysis of tryptic peptides obtained from native parvalbumin and Uniprot database matching, showed two isoforms of parvalbumin, named PV-I (99% coverage) and PV-II (75% coverage), respectively (data shown as Figure 2).After this insert sentences from lines 294-298 PV-I and PV-II are the most common parvalbumin isoforms, their protein structure are similar, both of them contain 108~109 amino acids, but the amino acid sequence is different, and the homology of sequence is only 50 to 80%. Both isoforms can cause allergic reactions, but PV-I has stronger binding ability with IgG and IgE than PV-II[38].

Response44: Thank you for pointing this out. It has been revised in the manuscript Line 266-267, Line 283-288, Line 294-298.

  1. Lines 289- 293 Replace products from with samples in line 289. This part move after sentences in lines 294-297.

Response45: Thank you for pointing this out. It has been revised in the manuscript.

  1. Replace N with n .Line 317, 318 Insert parvalbumin isoforms before PV-I, replace products from with samples. Line 332 A replace with a. Line 326 add et al. and year.Line 329 Replace marked text with Marchler-Baure et a. (2015)[Number of reference].Line 336 Immunogenicity.

Response46: Thank you for pointing this out. It has been revised in the manuscript.

47 Lines 343 Replace with Amino acid sequence of parvalbumin isoforms PV-I (a), and PV-II (b) in mandarin fish. Lines 339-341 Replace with Peptide matching diagram of parvalbumin isoforms PV-I (a), and PV-II (c) in native sample and their mass spectrograms (b) and (d), respectively.

Response47: Thank you for pointing this out. It has been revised in the manuscript.

48 Lines 343Replace with Amino acid sequence of parvalbumin isoforms PV-I (a), and PV-II (b) in mandarin fish. Lines 339-341Replace with Peptide matching diagram of parvalbumin isoforms PV-I (a), and PV-II (c) in native sample and their mass spectrograms (b) and (d), respectively.Lines 352

Replace marked part  with all samples obtained during both type of heating (microwave or induction cooker) were lower.When compare the inhibition rates of different sugars (samples G3, M2 and S1) to the inhibition rate of native parvalbumin, it was found that glucose had the highest inhibition effect, up to 66.03%,  followed maltose and sucrose.Line 357-359 Replace with It was found that the inhibition rates of samples obtained during microwave heating were significantly lower than those obtained during induction cooker heating.Replace There with These.

Response48: Thank you for pointing this out. It has been revised in the manuscript.

  1. Lines 375-377 The conclusion about digestibility G3 and G-Heating samples cannot clearly see from the figure. After 2 minutes there is no visible protein bands in both samples.

Response49: Thank you for your comment. Resistance of protein to pepsin digestion can be used as a marker of potential allergenicity. It means the easier it is digested, the lower the possibility of allergy. Figure 5 objectively shows the experimental results.

  1. Because reference 21 is not cited in the text, and order of some part of text was changed it should change reference number in the manuscript.

Response50: Thank you for pointing this out. It has been revised in the manuscript.

Reviewer 4 Report

Thanks to the authors for accepting the suggestions. I have no more comments.

Best regards, 

Author Response

Dear  Reviewer,

Thanks very much for taking your time to review this manuscript. We really appreciate all your generous comments and suggestions!

Thanks for your kindness and best wishes.

Yours sincerely,

Dr. Hong Zhang
